# A randomized placebo-controlled clinical trial for pharmacological activation of BCAA catabolism in patients with type 2 diabetes

Froukje Vanweert [1], Michael Neinast[2], Edmundo Erazo Tapia[1], Tineke van de Weijer[1,3], Joris Hoeks [1], Vera B. Schrauwen-Hinderling [1,3], Megan C. Blair[2], Marc R. Bornstein [2], Matthijs K. C. Hesselink [1], Patrick Schrauwen [1], Zoltan Arany [2] & Esther Phielix [1✉]

Elevations in plasma branched-chain amino acid (BCAA) levels associate with insulin resistance and type 2 diabetes (T2D). Pre-clinical models suggest that lowering BCAA levels improve glucose tolerance, but data in humans are lacking. Here, we used sodium phenylbutyrate (NaPB), an accelerator of BCAA catabolism, as tool to lower plasma BCAA levels in patients with T2D, and evaluate its effect on metabolic health. This trial (NetherlandsTrialRegister: NTR7426) had a randomized, placebo-controlled, double-blind cross-over design and was performed in the Maastricht University Medical Center (MUMC+), the Netherlands, between February 2019 and February 2020. Patients were eligible for the trial if they were 40–75years, BMI of 25–38 kg/m², relatively well-controlled T2D (HbA1C < 8.5%) and treated with oral glucose-lowering medication. Eighteen participants were randomly assigned to receive either NaPB 4.8 g/m²/day and placebo for 2 weeks via controlled randomization and sixteen participants completed the study. The primary outcome was peripheral insulin sensitivity. Secondary outcomes were ex vivo muscle mitochondrial oxidative capacity, substrate oxidation and ectopic fat accumulation. Fasting blood samples were collected to determine levels of BCAA, their catabolic intermediates, insulin, triglycerides, free fatty acids (FFA) and glucose. NaPB led to a robust 27% improvement in peripheral insulin sensitivity compared to placebo (ΔRd:13.2 ± 1.8 vs. 9.6 ± 1.8 µmol/kg/min, $p = 0.02$). This was paralleled by an improvement in pyruvate-driven muscle mitochondrial oxidative capacity and whole-body insulin-stimulated carbohydrate oxidation, and a reduction in plasma BCAA and glucose levels. No effects were observed on levels of insulin, triglycerides and FFA, neither did fat accumulation in muscle and liver change. No adverse events were reported. These data establish the proof-of-concept in humans that modulating the BCAA oxidative pathway may represent a potential treatment strategy for patients with T2D.

[1] Department of Nutrition and Movement Sciences, NUTRIM School of Nutrition and Translational Research in Metabolism, Maastricht University, Maastricht 6229 ER, The Netherlands. [2] Cardiovascular Institute, Perelman School of Medicine, University of Pennsylvania, Philadelphia, Pennsylvania, PA 19104, USA. [3] Department of Radiology and Nuclear Medicine, Maastricht University Medical Center, Maastricht 6229 ER, The Netherlands. ✉email: esther.phielix@maastrichtuniversity.nl

In the past three decades, the prevalence of type 2 diabetes (T2D) has risen dramatically and developed into a major global health problem[1]. Extensive research has shown that T2D is a multifactorial disease characterized by insulin resistance accompanied with a broad scale of tissue-specific and whole-body metabolic disturbances, such as low mitochondrial function, metabolic inflexibility and ectopic lipid accumulation[2–4]. In recent years, several observational studies, including work of our own, identified elevated plasma branched-chain amino acids (BCAA) levels in obese people and patients with T2D, associating with insulin resistance[5–8]. A rise of BCAA plasma levels may even predict the onset of T2D[9,10]. Why BCAA levels accumulate in plasma is currently unknown, but recent data -predominantly obtained from animal models- hypothesize that levels accumulate through suppression of the BCAA-catabolic pathway[11–14].

BCAA catabolism involves initial transamination of BCAA to branched-chain α-keto acids (BCKAs) by the BCAA aminotransferase (BCAT), followed by decarboxylation of BCKAs by the BCKA dehydrogenase complex (BCKD), the rate limiting enzyme of BCAA catabolism[11]. The latter complex is activated via dephosphorylation by the PPM1K phosphatase, and inactivated via phosphorylation by the BCKD kinase. There is evidence that BCKD kinase activity increases with the progression of insulin resistance and T2D, resulting in reduced BCAA oxidation and a subsequent rise of BCAA levels in plasma[11,13,15,16]. In line with this hypothesis, we recently reported lower whole-body leucine oxidation rates in patients with T2D compared to healthy control participants[8].

Several rodent studies have compellingly demonstrated that promoting BCAA oxidation benefits glucose metabolism and alleviates insulin resistance. Administration of the compound BT2 (3,6-dichlorobenzothiophene-2-carboxylic acid), a potent and specific inhibitor of the BCKD kinase[17], in mice accelerated BCAA oxidation and reduced plasma BCAA levels[18–23]. As a result, hepatic steatosis decreased and glucose disposal in peripheral tissues improved. Administration of this BCKD kinase inhibitor furthermore attenuated insulin resistance in high-fat diet-induced obese mice[24]. These studies provide proof-of-concept evidence for the therapeutic potential of manipulating BCAA metabolism[24], and raise the question if this strategy may form a treatment strategy in patients with T2D. BT2 binds BCKD kinase in an allosteric pocket, leading to inhibition of kinase activity. BT2 is not suitable for human use, but sodium phenylbutyrate (NaPB), an FDA approved drug regularly prescribed in patients suffering from urea cycle disorders, binds the same allosteric pocket inhibiting BCKD kinase[17], and lowers plasma BCAA levels in humans[25].

In the present study, NaPB was administered 'off-label' as add-on medication to patients with T2D as a tool to lower BCAA plasma levels. We evaluated a broad range of metabolic parameters after a 2-week intervention period, and compared results to a placebo arm. We hypothesize that NaPB treatment effectively decreases BCAA plasma levels and improves patients' metabolic health, including peripheral insulin sensitivity, muscle mitochondrial oxidative capacity, whole-body substrate oxidation and decreases ectopic fat accumulation in muscle and liver.

## Results

**Experimental design.** The study had a randomized, double-blind, placebo-controlled, crossover design (Fig. 1). Sixteen participants underwent 2 intervention arms, which involved daily administration of 4.8 g/m²/d NaPB or placebo, separated by a washout period of 6 to 8 weeks (Fig. 2). After 2 weeks of each treatment, participants underwent comprehensive metabolic evaluation. The primary outcome was insulin sensitivity, measured by a the 2-step euglycemic/hyperinsulinemic. Secondary outcomes were ex vivo mitochondrial oxidative capacity in skeletal muscle, measured with high-resolution respirometry, whole-body substrate oxidation, assessed with indirect calorimetry, and ectopic fat accumulation in muscle and liver measured with proton magnetic resonance spectroscopy ($^1$H-MRS).

**Baseline characteristics and treatment compliance.** Baseline characteristics are reported in Table 1. Compliance was determined by weighing the medication granules and by analysis of concentrations of phenylbutyrate and phenylacetylglutamine in plasma at the end of the treatment periods. Compliance rate (ratio taken dose/prescribed dose) in the NaPB arm was $95.8 \pm 13.8\%$ and $95.4 \pm 11.0\%$ for placebo. Concentrations of phenylbutyrate and phenylacetylglutamine in plasma were significantly higher in the NaPB arm compared to placebo ($p < 0.0001$, Suppl. Fig. 1), which together confirms compliance to the treatment intervention.

**NaPB treatment improved peripheral insulin sensitivity and whole-body carbohydrate oxidation.** NaPB treatment improved whole-body insulin sensitivity, as assessed by a 2-step hyperinsulinemic-euglycemic clamp. The change in hyperinsulinemic-stimulated glucose disposal rate ($\Delta Rd$), robustly improved by 27% ($p = 0.02$) after NaPB treatment compared to placebo (Fig. 3a, Table 2). During the low-insulin phase of the clamp, insulin-suppressed EGP did not change ($p = 0.84$, Table 2), but EGP became 6% more suppressed during the high-insulin phase with NaPB compared to placebo ($p = 0.02$, Fig. 3b, Table 2). Plasma FFA levels were suppressed to a similar extent between NaPB and placebo during both the low and high-insulin phases of the clamp ($p = 0.57$ and $p = 0.84$, respectively, Table 2). These data suggest that NaPB treatment specifically improves muscle insulin sensitivity, with a significant, albeit modest, improvement of hepatic insulin sensitivity upon high insulin concentration.

Whole-body carbohydrate, fat and protein oxidation measured during the basal and low-insulin phase, remained similar between NaPB and placebo groups (Table 2). During high-insulin phase, carbohydrate oxidation was 10% higher after NaPB treatment ($p = 0.03$, Fig. 3c, Table 2), while fat and protein oxidation did not change ($p = 0.16$ and $p = 0.45$, Table 2). The change in insulin-stimulated non-oxidative glucose disposal ($\Delta NOGD$) was similar between conditions ($p = 0.46$, Table 2). In addition, metabolic flexibility, expressed as the change from basal respiratory exchange ratio (RER) to insulin-stimulated RER, tended to improve under high insulin conditions with NaPB vs. placebo ($p = 0.07$, Fig. 3d, Table 2).

**NaPB treatment elevates muscle mitochondrial oxidative capacity.** Mitochondrial oxidative capacity, measured in permeabilized muscle fibers was higher after NaPB treatment compared to placebo. In the presence of pyruvate, a carbohydrate-derived substrate, ADP-driven state 3 respiration with parallel electron input to complex II (malate + glutamate + succinate), significantly improved by 10% (Fig. 3e, $p = 0.04$). NaPB treatment tended to improve mitochondrial oxygen consumption upon the stimulation of complex I (malate + glutamate) (Suppl. Table 1, $p = 0.07$), without differences found in the maximal respiratory capacity upon the chemical uncoupler FCCP (Suppl. Table 1, $p = 0.12$). In contrast, ADP-driven state 3 respiration fueled by the lipid-derived substrate octanoyl-carnitine, did not change ($p = 0.25$, Fig. 3f). Also, no differences were observed for other respiratory states in the presence of octanoyl-carnitine, as shown in Suppl. Table 1. Together, these respiratory data in permeabilized skeletal muscle fibers indicate an improvement in the capacity for the oxidation of carbohydrate-derived substrates. The

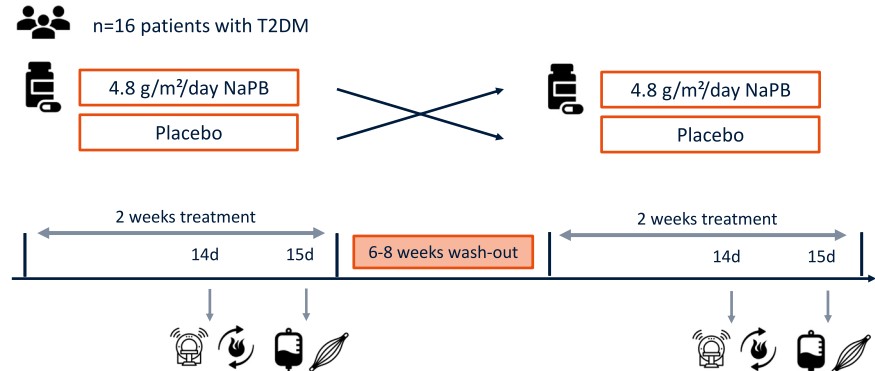

**Fig. 1 Experimental design.** In this crossover study, participants were randomly assigned to start with 2-week NaPB supplementation or placebo treatment. After a washout period of 6–8 weeks, participants switched from intervention arm such that all participants served as their own control. In each treatment arm, measurements were performed after 2 weeks treatment, including magnetic resonance spectroscopy (day 14), whole-body 24 h energy metabolism and substrate oxidation (day 14), 2-step euglycemic hyperinsulinemic clamp (day 15) and muscle biopsies (day 15) were taken. T2DM patients with type 2 diabetes mellitus, NaPB sodium phenylbutyrate.

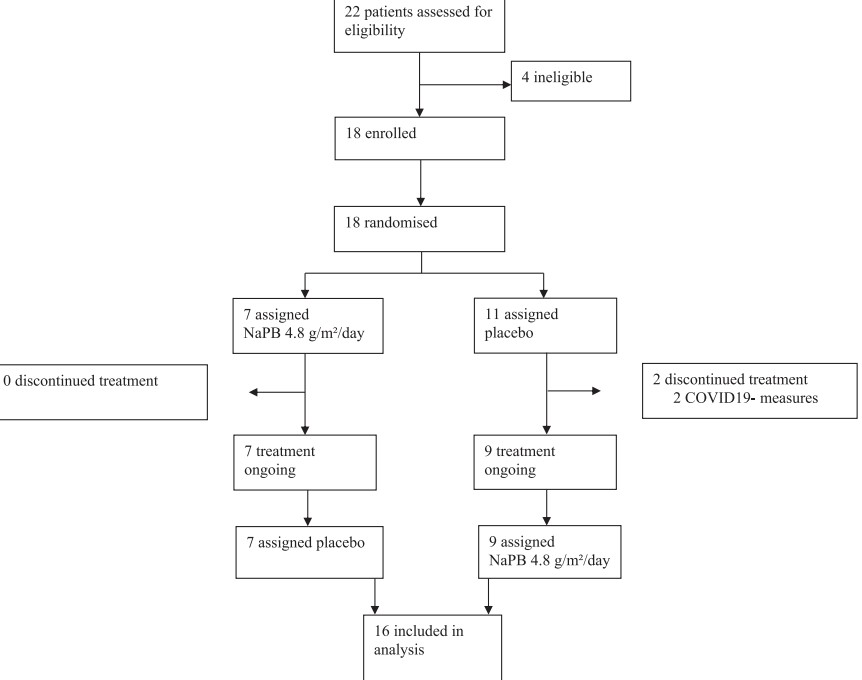

**Fig. 2 Trial profile.** Sixteen participants completed the treatment and were included in analysis. Two patients discontinued treatment prematurely due to COVID19-measures.

complete data set for the different respiratory states are reported in Suppl. Table 1.

**NaPB treatment did not alter ectopic lipid storage.** $^1$H-MRS was applied in the m.tibialis anterior and liver to measure in vivo lipid content. NaPB treatment had no effect on IMCL content (NaPB: $0.61 \pm 0.08\%$ vs. placebo: $0.50 \pm 0.06\%$; $p = 0.14$), or IHL content (NaPB: $13.5 \pm 3.1\%$ vs. placebo: $11.7 \pm 2.4\%$; $p = 0.20$, Suppl. Table 2). Further analysis of hepatic lipid composition revealed no effect of NaPB compared to placebo (%PUFA: $13.15 \pm 1.49$ vs $14.09 \pm 1.39$, $p = 0.50$; %MUFA: $42.93 \pm 1.79$ vs. $41.70 \pm 1.62$, $p = 0.51$; %SFA: $43.91 \pm 1.29$ vs. $44.21 \pm 1.67$, $p = 0.88$, Suppl. Table 2).

**NaPB treatment reduces plasma BCAA levels.** As hypothesized, 2 weeks NaPB treatment resulted in 8% lower total BCAA levels

($p = 0.03$, Fig. 4a) compared to placebo, with a significant decline of all three individual BCAA ($-10\%$ for valine, $p = 0.009$; $-7\%$ for leucine, $p = 0.03$; $-6\%$ for isoleucine, $p = 0.05$; Fig. 4b–d, respectively). The complete amino acids profile after both treatment arms are presented in Suppl. Table 3. In addition, fasting glucose levels tended to be lower after NaPB treatment compared to placebo ($7.7 \pm 0.4$ mmol/L vs. $8.2 \pm 0.5$ mmol/L, $p = 0.06$, Table 3), without any effects observed for insulin, triglycerides and FFA (Table 3).

**Effect of NaPB on BCAA catabolism-related metabolites in plasma.** Metabolomic analysis in plasma showed a significant decrease of α-ketoisovalerate (KIV), the valine-derived BCKA ($p = 0.02$, Fig. 3g) with NaPB treatment compared to placebo. Leucine- and isoleucine-derived BCKA's, (α-ketoisocaproate (KIC) and α-keto β-methylvalerate (KMV); not distinguishable by

**Table 1 Baseline characteristics[a].**

|  | Mean ± SD |
|---|---|
| Gender, n (F/M) | 3/13 |
| Age, years | 66 ± 6 |
| Body weight, kg | 90.8 ± 15.4 |
| Height, cm | 174.7 ± 8.1 |
| BMI, kg/m$^2$ | 29.6 ± 3.3 |
| Fasting glucose, mmol/L | 8.4 ± 1.5 |
| HbA1c, % | 6.5 ± 0.6 |
| ASAT, U/L | 23.8 ± 3.9 |
| ALAT, U/L | 30.3 ± 7.7 |
| GGT, U/L | 32.6 ± 14.6 |
| Potassium (mmol/L) | 4.5 ± 0.1 |
| Creatinine (μmol/L) | 84.8 ± 4.8 |
| Bilirubin | 12.1 ± 2.8 |
| Hemoglobin (mmol/L) | 8.8 ± 0.2 |
| Oral glucose lowering medication, n | 16 |
| Metformin only | 8 |
| Sulphonylurea derivates only | 2 |
| Metformin + sulphonylurea derivates | 6 |

[a]n = 16. Screening values are means ± SD BMI body mass index, ASAT aspartate aminotransferase, ALAT alanine aminotransferase, GGT gamma glutamyltransferase.

mass spectroscopy) did not change ($p = 0.12$, Fig. 3h), however trended in the same direction as KIV. The reduction of plasma BCAA levels and KIV reflects reduced accumulation of plasma substrates upstream of the BCKD-complex. 3-hydroxyisobutyrate (3-HIB) is an intermediate of valine catabolism downstream of the BCKD complex, and is unique in its ability to escape the mitochondria and appear in the plasma. Interestingly, 3-HIB significantly decreased in the NaPB treatment arm compared to placebo ($p = 0.02$, Fig. 3i). The change in 3-HIB concentrations between placebo and NaPB arms negatively correlated with the change in whole-body carbohydrate oxidation ($r = -0.55$, $p = 0.05$), i.e. the subjects with highest NaPB-improved whole body carbohydrate oxidation showed the largest decrease in plasma 3-HIB, suggesting that the decrease of 3-HIB in plasma after NaPB treatment reflects improved mitochondrial TCA flux. Consistent with our findings, elevated levels of 3-HIB were previously shown to be higher under insulin resistant conditions[21,26,27]. In contrast, we did not found associations between 3-HIB concentrations and other measures for insulin sensitivity, like Rd ($r = 0.08$, $p = 0.76$), EGP ($r = 0.13$, $p = 0.64$) and fasting glucose values ($r = 0.27$, $p = 0.31$).

**No change in sleeping metabolic rate and nocturnal substrate oxidation with NaPB treatment**. Sleeping metabolic rate, measured during an overnight stay in the respiration chamber, was not affected by NaPB treatment compared to placebo ($7.0 ± 0.3$ MJ/d vs. $7.1 ± 0.2$ MJ/d, respectively, $p = 0.92$, Suppl. Table 4). In addition, RER and substrate oxidation during the night were similar between the two conditions (Suppl. Table 4).

**Unchanged body composition upon treatment arms**. Two weeks of NaPB treatment did not affect body composition. Percentage fat free mass (NaPB: $63.6 ± 2.1\%$ vs. placebo: $63.4 ± 2.0 \%$, $p = 0.79$) and fat mass (NaPB: $36.1 ± 2.2\%$ vs. placebo: $36.6 ± 2.0\%$, $p = 0.50$) remained similar. In line, no effect of NaPB was observed for the change in total body weight (NaPB: $-0.01 ± 0.5$ kg vs. placebo: $-0.19 ± 0.49$ kg, p = 0.56).

## Discussion

Recent metabolomics and comprehensive metabolic profiling studies, including work of our own, consistently show elevated BCAA plasma levels in obese/T2D rodent models as well as in patients with T2D[5,9,10,28–30]. Here, we prescribed NaPB 'off-label' to patients with T2D to stimulate the oxidation of BCAA aiming to lower its systemic concentrations. We show that 2 weeks of NaPB treatment effectively reduced plasma BCAA levels. This reduction was accompanied by a 27% improved peripheral glucose disposal, mainly exerted by enhanced insulin-stimulated carbohydrate oxidation. In addition, NaPB treatment increased ex vivo mitochondrial oxidative capacity upon pyruvate in muscle by 10%. These data provide evidence in humans that pharmacologically boosting BCAA oxidation, lowers BCAA plasma levels in patients with T2D and results in beneficial outcomes on patients' glucose metabolism.

NaPB-enhanced BCAA catabolism resulted in improved peripheral insulin sensitivity, which mainly involved glucose uptake by muscle, accompanied by a tendency towards lower fasting plasma glucose levels. The improved insulin-stimulated glucose uptake was attributed to enhanced glucose oxidation, without changes in NOGD, the latter reflecting glycogen synthesis. These findings align with cell and rodent studies, in which the BCKD kinase inhibitor BT2, like NaPB, effectively improved glucose tolerance of peripheral tissues, attenuated insulin resistance and enhanced glucose oxidation via stimulated insulin signaling in high-fat diet-induced obese mice. We found that NaPB treatment did not show major effects on hepatic insulin sensitivity, which suggest that insulin resistance in the liver is less responsive to NaPB treatment. Together, our results show that the effects of NaPB treatment in patients with T2D for a great part take place in peripheral tissues, mainly muscle, which matches with the observation that skeletal muscle in humans has the highest capacity for BCAA catabolism[31,32].

We found a decrease in 3-HIB plasma concentrations with NaPB treatment, paralleled by improved insulin sensitivity and furthermore associated with improved whole-body carbohydrate oxidation. This finding aligns with observational human studies, which showed that elevated 3-HIB plasma levels associate with insulin resistance and risk of incident T2D[21,27]. 3-HIB is formed from valine breakdown and becomes hydrolyzed by 3-HIB-CoA hydrolase, whereafter it can leave the mitochondria and cell to the extracellular fluid or plasma[33]. As NaPB boosts BCAA catabolism, the decrease in 3-HIB plasma levels could be a resultant of better mitochondrial TCA cycling. Alternatively, oxidation of BCAA may have been repartitioned to tissues less amenable to 3-HIB secretion.

The obtained metabolic results are consistent in showing improved glucose handling at different levels: enhanced insulin-stimulated glucose disposal rate, improved insulin-stimulated whole-body carbohydrate oxidation and tendency towards a higher RER, as well indirectly, by increased mitochondrial oxidative capacity for pyruvate oxidation in muscle fibers. The mechanistic link between elevated plasma BCAA levels and insulin resistance remains poorly understood, although several hypotheses have been proposed[14,34]. Our data support that the lowering of BCAA plasma levels, and the concomitant reduction in plasma BCKA levels, may alleviate the inhibition on insulin signaling and glucose oxidation, leading to improved insulin sensitivity. A hypothesis is that the NaPB-induced improvement in insulin sensitivity could be explained by the reduced activation of the mammalian target of rapamycin complex (mTOR)[14], as elevated plasma BCAA levels are thought to impair insulin signaling via activation of S6 kinase (p70S6K) and mTOR[35–38]. In addition, previous publications performed in animal heart tissue collectively showed that the accumulation of BCAA levels and its derived metabolites inhibit PDH activity, thereby hampering glucose oxidation and insulin sensitivity[39,40]. Therefore, NaPB-

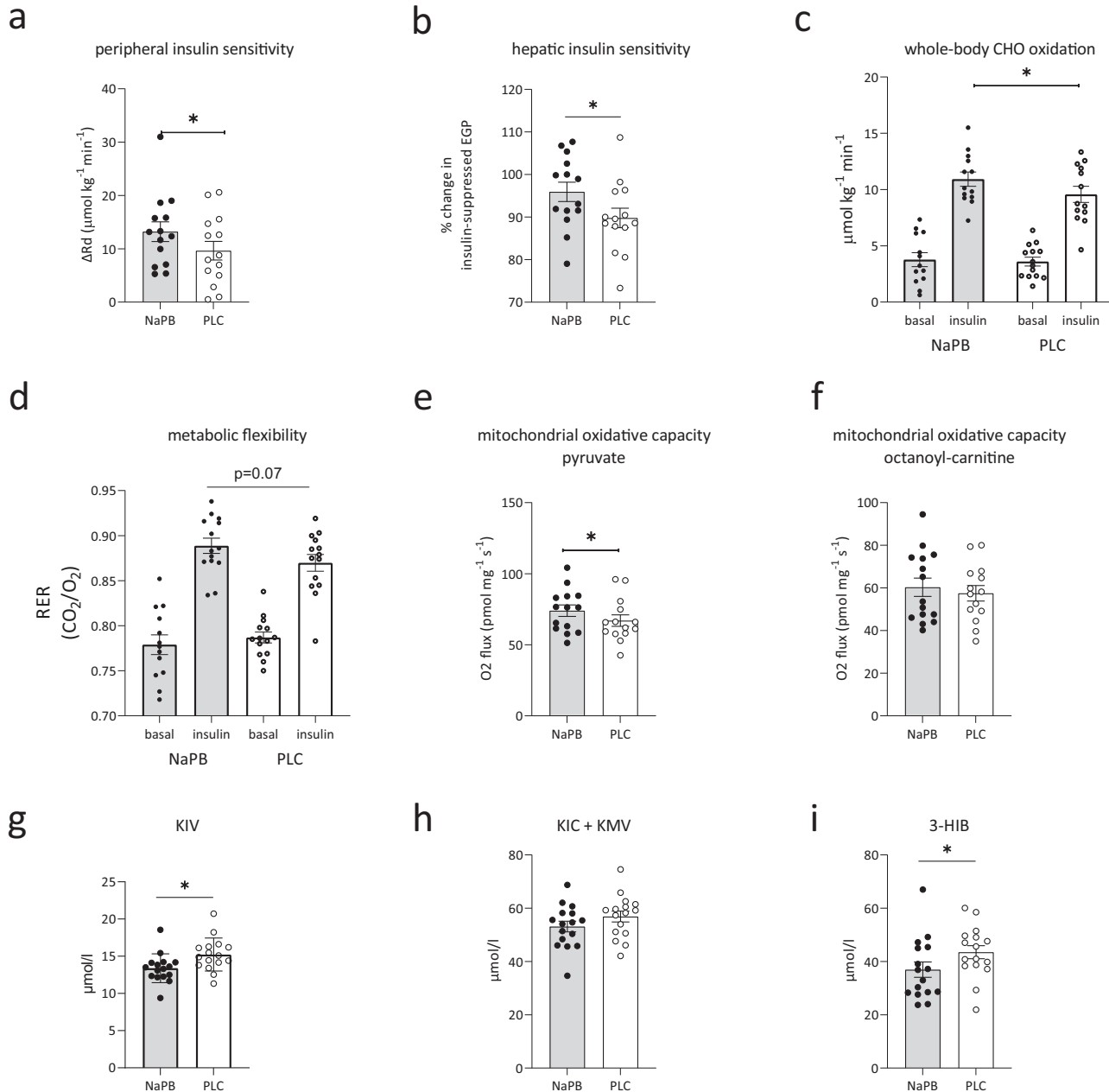

**Fig. 3 NaPB treatment effects on metabolic read-out parameters and plasma metabolites.** Metabolic parameters measured in patients with T2D after 2-week treatment with NaPB (gray bars, $n = 16$) and placebo (white bars, $n = 16$). **a** peripheral insulin sensitivity expressed as the change in insulin-stimulated Rd ($\mu$mol kg$^{-1}$ min$^{-1}$, $p = 0.02$), calculated as the difference between Rd under high-insulin infusion and Rd during basal conditions, **b** hepatic insulin sensitivity expressed as the change in insulin-suppressed EGP (%, $p = 0.02$) under basal conditions versus high insulin infusion, **c** carbohydrate oxidation ($\mu$mol kg$^{-1}$ min$^{-1}$, $p = 0.03$) under basal and high insulin infusion, and **d** metabolic flexibility, expressed as change in RER from the basal to the high-insulin infusion ($p = 0.07$), skeletal muscle ex vivo mitochondrial respiratory capacity **e** upon carbohydrate-derived substrate pyruvate (pmol mg$^{-1}$ s$^{-1}$, $p = 0.03$), and **f** upon lipid-derived substrate octanoyl carnitine (pmol mg$^{-1}$ s$^{-1}$, $p = 0.25$) with parallel electron input to complex II (malate + glutamate + succinate), Fasting plasma metabolites measured after 2-week treatment with NaPB: **g** fasting total KIV levels ($\mu$mol/l, $p = 0.02$), **h** fasting KIC + KMV levels ($\mu$mol/l, $p = 0.12$) and **i** fasting concentration of 3-HIB ($\mu$mol/l, $p = 0.02$). Data are expressed as mean ± SE. The intervention effect was analyzed using the paired student t-test. *$P < 0.05$. BCAA branched-chain amino acids, NaPB sodium-phenylbutyrate, T2D type 2 diabetes, PLC placebo, Rd glucose disposal, EGP endogenous glucose production, RER respiratory exchange ratio, 3-HIB 3-hydroxyisobutyrate, KIC α-ketoisocaproate, KIV α-ketoisovalerate, KMV α-keto-β-methylvalerate. Source data are provided as a Source Data file.

induced improved glucose oxidation could also be explained by higher PDH activity.

Previously we reported 60% improved insulin sensitivity and 33% improved mitochondrial oxidative capacity after a 3 months progressive exercise training program in patients with T2D[41,42]. Exercising is by far the most effective strategy to reduce diabetes-related metabolic disturbances, as well prevent or delay the onset. The 27% improvement in insulin sensitivity we observed in the current study is therefore quite significant, amounting to about half of that achieved by exercise. The results emphasize the relevance of BCAA catabolism in insulin resistance in humans, as well the potential impact of this treatment strategy on metabolic

**Table 2 NaPB treatment improved peripheral insulin sensitivity and whole-body carbohydrate oxidation[a].**

| | NaPB | Placebo | P value |
|---|---|---|---|
| **Ra ($\mu$mol · kg$^{-1}$ · min$^{-1}$) [b]** | | | |
| Baseline | 10.6 ± 0.7 | 12.1 ± 1.1 | 0.19 |
| Low insulin | 10.5 ± 0.7 | 10.6 ± 0.5 | 0.81 |
| High insulin | 23.4 ± 2.4 | 21.1 ± 2.2 | 0.01* |
| $\Delta$ baseline - low | −0.1 ± 0.5 | −1.5 ± 0.9 | 0.32 |
| $\Delta$ baseline – high | 11.2 ± 2.0 | 7.9 ± 1.9 | 0.01* |
| **Rd ($\mu$mol · kg$^{-1}$ · min$^{-1}$) [b]** | | | |
| Baseline | 10.9 ± 0.9 | 12.2 ± 1.1 | 0.11 |
| Low insulin | 10.6 ± 0.8 | 10.6 ± 0.6 | 0.92 |
| High insulin | 24.1 ± 2.3 | 21.9 ± 2.1 | 0.01* |
| $\Delta$ low ins - baseline | −0.3 ± 0.7 | −1.4 ± 0.7 | 0.23 |
| $\Delta$ high ins - baseline | 13.2 ± 1.8 | 9.6 ± 1.8 | 0.02* |
| **EGP ($\mu$mol · kg$^{-1}$ · min$^{-1}$) [c]** | | | |
| Baseline | 10.6 ± 0.9 | 12.1 ± 1.1 | 0.19 |
| Low insulin | 5.9 ± 0.4 | 6.4 ± 0.6 | 0.17 |
| High insulin | 1.0 ± 0.2 | 1.8 ± 0.3 | 0.02* |
| % suppression low vs baseline | 49.2 ± 2.3 | 49.6 ± 2.6 | 0.84 |
| % suppression high vs baseline | 95.9 ± 2.3 | 89.9 ± 2.3 | 0.02* |
| **NOGD ($\mu$mol · kg$^{-1}$ · min$^{-1}$) [c]** | | | |
| Baseline | 7.3 ± 0.8 | 8.1 ± 1.2 | 0.54 |
| Low insulin | 3.6 ± 0.6 | 4.0 ± 0.7 | 0.70 |
| High insulin | 11.8 ± 1.6 | 11.8 ± 1.5 | 0.99 |
| $\Delta$ high ins - baseline | 3.3 ± 1.4 | 3.2 ± 1.4 | 0.46 |
| **Carbohydrate oxidation ($\mu$mol · kg$^{-1}$ · min$^{-1}$) [b]** | | | |
| Baseline | 3.8 ± 0.6 | 3.6 ± 0.4 | 0.57 |
| Low insulin | 6.6 ± 0.7 | 6.6 ± 0.6 | 0.97 |
| High insulin | 10.9 ± 0.6 | 9.6 ± 0.7 | 0.03* |
| **Fat oxidation ($\mu$mol · kg$^{-1}$ · min$^{-1}$) [b]** | | | |
| Baseline | 2.6 ± 0.1 | 2.6 ± 0.2 | 0.57 |
| Low insulin | 2.2 ± 0.2 | 2.3 ± 0.2 | 0.75 |
| High insulin | 1.5 ± 0.1 | 1.7 ± 0.2 | 0.16 |
| **Protein oxidation ($\mu$mol · kg$^{-1}$ · min$^{-1}$) [b]** | | | |
| Baseline | 6.8 ± 0.5 | 6.9 ± 0.6 | 0.77 |
| Low insulin | 4.4 ± 0.4 | 4.4 ± 0.5 | 0.84 |
| High insulin | 4.0 ± 0.2 | 4.5 ± 0.6 | 0.45 |
| **Plasma FFA's ($\mu$mol/L) [b]** | | | |
| Baseline | 612 ± 37 | 628 ± 31 | 0.52 |
| Low insulin | 212 ± 20 | 226 ± 20 | 0.31 |
| High insulin | 89 ± 15 | 92 ± 12 | 0.67 |
| % suppression low insulin | 65 ± 3 | 64 ± 3 | 0.57 |
| % suppression high insulin | 85 ± 3 | 85 ± 2 | 0.84 |
| **Respiratory exchange ratio [c]** | | | |
| Baseline | 0.78 ± 0.01 | 0.78 ± 0.01 | 0.66 |
| Low insulin | 0.83 ± 0.01 | 0.82 ± 0.01 | 0.94 |
| High insulin | 0.89 ± 0.01 | 0.87 ± 0.01 | 0.07 |
| $\Delta$ high ins - baseline | 0.11 ± 0.01 | 0.10 ± 0.01 | 0.37 |

[a]Data expressed as mean ± SE. The intervention effect was analyzed using the paired student t-test. *P values < 0.05 NaPB vs. Placebo. EGP endogenous glucose production, FFA free fatty acids, NOGD nonoxidative glucose disposal, NaPB sodium phenylbutyrate, Ra rate of glucose appearance, Rd rate of glucose disappearance.
[b]n = 14,
[c]n = 15.

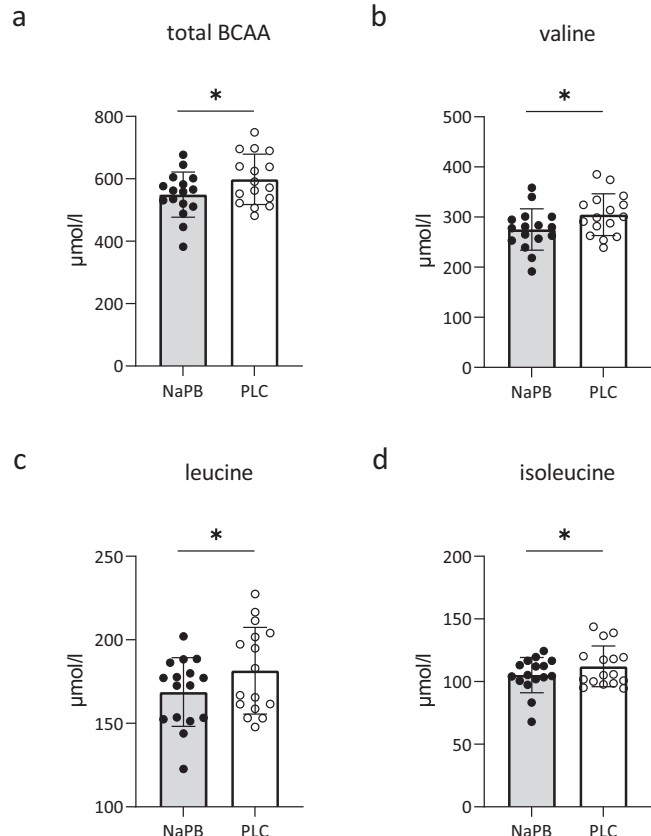

**Fig. 4 NaPB treatment reduces plasma BCAA levels.** Fasting plasma BCAA's were measured in patients with T2D after 2-week treatment with NaPB (gray bars, n = 16) and placebo (white bars, n = 16). **a** Total BCAA levels ($\mu$mol/l, p = 0.03), **b** valine levels ($\mu$mol/l, p = 0.009) and **c** leucine values ($\mu$mol/l, p = 0.03), and **d** isoleucine ($\mu$mol/l, p = 0.05). Data are expressed as mean ± SE. The intervention effect was analyzed using the paired student t-test. *P < 0.05. NaPB sodiumphenylbutyrate, PLC placebo, T2D type 2 diabetes. Source data are provided as a Source Data file.

**Table 3 NaPB treatment reduces fasting glucose levels [a].**

| Parameter | NaPB | Placebo | P value |
|---|---|---|---|
| Glucose, mmol/L [b] | | | |
| day 0 | 8.7 ± 0.5 | 8.5 ± 0.5 | 0.53 |
| day 15 | 7.7 ± 0.4 | 8.2 ± 0.5 | 0.06 |
| $\Delta$ day 0-day 15 | −1.0 ± 0.2 | −0.3 ± 0.3 | 0.09 |
| Insulin, mU/L | 10.9 ± 1.2 | 11.1 ± 1.4 | 0.78 |
| Triglycerides, mmol/L | 1.9 ± 0.2 | 1.8 ± 0.2 | 0.50 |
| Free fatty acids, $\mu$mol//L | 614 ± 42 | 632 ± 50 | 0.51 |

[a]n = 16. Data expressed as mean ± SE. The intervention effect was analyzed using the paired student t-test. *P values < 0.05 NaPB vs. Placebo. Fasting blood samples were taken after 2 weeks (15 days) of NaPB treatment or placebo after an overnight fast. NaPB sodium phenylbutyrate, BCAA branched-chain amino acids.
[b]n = 15.

health in metabolically compromised people. With this short duration time we observed these significant improvements, as well as tendencies for decreasing fasting glucose values and improved metabolic flexibility, which may form the lead for future studies with longer treatment duration.

Our study has limitations. We recognize unequal gender distribution. Although there were no restrictions for females participating in the study, a majority of male patients subscribed to the study. Therefore, future studies are needed to investigate whether similar effects occur in both sexes. Participants' wide range of BMI (25–38 kg/m$^2$) could influence metabolic responses. However, after adjustment for BMI, p-values remained significant meaning that it is unlikely BMI affected the metabolic responses seen. Due to the invasive character of the study, we prioritized evaluating effects on skeletal muscle and liver. Reports, however, highlight the role of BCAA catabolism in adipose tissue[43,44]. Therefore, it would be of interest to study NaPB treatment effects

in human adipocytes and investigate its contributing effect on defining plasma BCAA levels. NaPB was given as add-on treatment, combined with oral antidiabetic agents of various mechanisms of action., More-over, 14 patients were on metformin treatment throughout the study, and 8 patients received metformin only/or a combined therapy with sulphonylurea derivates. Recently, it has been shown that metformin does not alter BCAA plasma levels[45], while the effect of sulphonylureas on BCAA metabolism or BCAA plasma levels have not yet been investigated. It would be of interest to perform sub-group analysis comparing two groups of patients' medication in order to investigate drug-drug interactions, however, our sample size was too small to de reliable sub-group analysis. Therefore, we cannot conclude to what extent the effects of NaPB depend on the co-medication given, which should be investigated in future trials.

In summary, the present randomized double-blind placebo-controlled trial shows that NaPB treatment decreases BCAA levels together with an improvement in peripheral insulin sensitivity and muscle mitochondrial oxidative capacity on pyruvate in patients with T2D. These findings demonstrate in humans that pharmacologically boosting BCAA catabolism exerts substantial beneficial effects on glucose homeostasis in patients with T2D, as has previously been shown in numerous rodent models. Our work strongly justifies future efforts to investigate this potential treatment strategy for this prevalent and debilitating disease.

## Methods

**Clinical study design.** Participants were enrolled between February 2019 and September 2019 at the Maastricht University Medical Center (MUMC+), the Netherlands, and the last subject completed in February 2020. Two dropouts were reported during the study (Fig. 2). The protocol was reviewed and approved by the Medical Ethical Review Committee of the MUMC + (Netherlands Trial Register ID: NTR7426) and conducted in accordance with the declaration of Helsinki. All participants were informed about the nature and risk of the experimental procedures before their written informed consent was obtained.

**Participants.** Sixteen male and postmenopausal females diagnosed with T2D for at least 1.5 years, participated in the study. Participants underwent a medical screening to check eligibility. Inclusion criteria were 40–75 y of age, BMI of 25–38 kg/m², relatively well-controlled T2D (HbA1C < 8.5%) treated with oral glucose lowering medication (metformin only, or in combination with sulphonylurea agents and/or DPPIV inhibitors) or drug naive for at least 3 months prior to the onset of the study. Patients had no signs of active cardiovascular diseases, liver or renal insufficiency. Exclusion criteria were unstable body weight (i.e. weight gain or loss > 5 kg in the last three months), participation in physical activity ≥ 3 times a week, insulin treatment, and MRI contra-indications.

**Experimental design.** The study had a randomized, double-blind, placebo-controlled, crossover design (Fig. 2). Each participant underwent 2 intervention arms, which involved daily administration of 4.8 g/m²/d NaPB or placebo. The participants were randomly assigned to receive either the NaPB or the placebo treatment, separated by a washout period of 6 to 8 weeks via controlled randomization. After 2 weeks, all participants underwent several measurements to evaluate patients' metabolic health. Three days before the start of these measurements, participants were instructed to refrain from strenuous physical activities and to continue their antidiabetic medication with the last dose taken on the evening before the hyperinsulinemic-euglycemic clamp test. Throughout the study, patients were asked to maintain their habitual diet and regular physical activity pattern.

**Study medication.** The study medication Pheburane (Lucane Pharma, Paris, France provided by Eurocept International, Ankeveen, The Netherlands) contained 483 mg/g NaPB and inactive ingredients (sucrose, maize starch, sodium, Hypromellose, ethylcellulose N7, macrogel 1500 and povidone K25). The placebo (produced by Tiofarma, Oud-Beijerland, the Netherlands) only contained the inactive ingredients of Pheburane. The daily dose of 4.8 g/m²/day (NaPB and placebo) was below the minimal, clinically prescribed (9.9–13.0 g/m²/day NaPB), to prevent reaching the maximally allowed dose of 20g/day, due to high body surface area of the overweight/obese participants and the development of unwanted side effects. The study medication was administered in the form of granules, taken orally 3 times a day, divided into 3 equal amounts and given with breakfast, lunch and dinner. The granules could be directly swallowed with a drink (e.g. water, fruit juices) or sprinkled on solid foods (e.g. mashed potatoes, yoghurt). Administration

of 4.8 g/m²/day NaPB was well tolerated and no adverse events or side effects were reported throughout the study.

**Overview of specified outcomes.** The primary outcome was peripheral insulin sensitivity, measured by the hyperinsulinemic-euglycemic clamps, expressed as the change in insulin-stimulated glucose disposal rate minus baseline (ΔRd).

Secondary outcomes were ex vivo mitochondrial oxidative capacity in skeletal muscle, measured with high-resolution respirometry expressed as $O_2$-flux, substrate oxidation, assessed with indirect calorimetry and fat accumulation in muscle and liver measured with proton magnetic resonance spectroscopy ($^1$H-MRS). Fasting blood samples were collected to determine levels of BCAA and their intermediates, insulin, triglycerides, FFA and glucose. In addition, phenylbutyrate levels were determined by LCMS to check compliance to the intervention.

**2-step hyperinsulinemic-euglycemic clamp.** A two-step hyperinsulinemic-euglycemic clamp with co-infusion of D-[6.6-$^2$H$_2$] glucose tracer (0.04 mg · kg$^{-1}$ · min$^{-1}$) started in the morning of day 15 at 06:30 to assess hepatic and whole-body insulin sensitivity. After a pre-infusion of D-[6.6-$^2$H$_2$] glucose tracer (0.04 mg/kg/min) for 3 h (basal phase), low dose insulin was infused at 10 mU · m$^{-2}$ · min$^{-1}$ for 3 h to assess hepatic insulin sensitivity (low insulin phase), with a subsequently raise in insulin concentration to 40 mU · m$^{-2}$ · min$^{-1}$ for 2.5 h (high insulin phase) to determine peripheral insulin sensitivity. Blood was frequently sampled to measure glucose concentration in arterialized blood. In addition, 20% glucose (enriched with D-[6.6-$^2$H$_2$] glucose tracer) was co-infused at a variable rate to maintain euglycemia (~6.0 mmol/L). During the last 30 min of each phase, blood samples were collected at 10 minutes interval to determine glucose tracer kinetics and indirect calorimetry was performed to measure substrate oxidation. Steele's single pool non-steady state equations were used to calculate the rate of glucose appearance (Ra) and disappearance (Rd)[46]. Volume of distribution was assumed to be 0.160 l/kg for glucose. The change in insulin-stimulated glucose disposal (ΔRd) was calculated by the difference between Rd measured under insulin-stimulated condition and basal conditions. Endogenous glucose production (EGP) was calculated as Ra minus exogenous glucose infusion rate. Hepatic insulin sensitivity was calculated as the percentage of EGP suppression during the low and high insulin phase. Nonoxidative glucose disposal (NOGD) was calculated as Rd minus carbohydrate oxidation, determined with indirect calorimetry. Isotopic enrichment of plasma glucose was determined by electron ionization gas chromatography-mass spectrometry as described previously[47].

**Indirect calorimetry.** Before and during the clamp test, indirect calorimetry was performed to measure energy expenditure and substrate utilization. Gas exchange was measured by open-circuit respirometry with an automated ventilated hood system (Omnical, Maastricht, the Netherlands) for 30 min. The Weir equation[48] was used to calculate whole-body resting energy expenditure from measurements of oxygen consumption and carbon dioxide production. Carbohydrate, fat and protein oxidation rates were calculated according to Frayn[49] and nitrogen was measured in 24 h collected urine samples.

**Skeletal muscle biopsies.** In the morning at day 15, before the start of the clamp test, a muscle biopsy was obtained from the m.vastus lateralis under local anesthesia (1% lidocaine without epinephrine), according to the technique of Bergström et al.[50]. A portion of muscle tissue was directly frozen in isopentane and stored at −80 °C until further analysis. Another portion was immediately placed in ice-cold preservation medium and processed for high resolution respirometry.

**High-resolution respirometry in permeabilized muscle fibers.** A small portion of the muscle biopsy sample was immediately placed in ice-cold biopsy preservation medium (BIOPS; OROBOROS Instruments, Innsbruck, Austria). Muscle fibers were permeabilized with saponin according to the technique of Veksler et al.[51]. After permeabilization, muscle fibers were transferred into ice-cold mitochondrial respiration buffer (MiRO5; OROBOROS Instruments). Subsequently, permeabilized muscle fibers (~2.5 g wet weight) were used for ex vivo high-resolution respirometry (Oxygraph, OROBOROS Instruments) by measuring oxygen consumption rate upon addition of several substrates. In every protocol applied, first, 4.0 mM malate was added to obtain state 2 respiration followed by addition of 1.0 mM octanoyl-carnitine or in presence of 5 mM pyruvate. In addition, 2 mM ADP with 10 mM glutamate was added to obtain ADP-driven state 3 respiration of complex I. Then 10 mM succinate was added to obtain state 3 respiration by activating both complex I and II. Finally, 1.0 mM carbonylcyanide p-trifluoromethoxyphenylhydrozone (FCCP) was added (in stepwise titration) to evaluate maximal respiratory capacity.

**Magnetic resonance spectroscopy: IHL and IMCL content.** On day 14, directly after the BodPod measurement, participants also underwent proton magnetic $^1$H-MRS to quantify intrahepatic lipid (IHL) and intramyocellular lipid (IMCL) content on a 3 T whole body scanner (Achieva 3T-X, Philips Healthcare, Best, the Netherlands). IHL and hepatic fatty acid composition was quantified as previously

described[52]. Values were corrected for T2 relaxation (T2 water: 26.3 ms and T2 $CH_2$: 57.8 ms) and given as ratios of $CH_2$ peak relative to the sum of $CH_2$ resonance and the unsuppressed water peak (in %). IMCL was measured in the m. tibialis anterior of the left leg, as previously described[53]. Values are given as T1- and T2-corrected ratios of the $CH_2$ peak[54] relative to the unsuppressed water peak (in %). Due to analytical problems only 13 participants could be included in the analyses of IMCL.

**Respiration chamber.** After the MRS measurements, in the late afternoon of day 14 of each intervention arm, participants consumed a standardized dinner before they went into the respiration chamber: a small room with a bed, toilet, TV and computer. During the overnight stay (for 12 hours) in this chamber, oxygen consumption and carbohydrate production were measured continuously in sampled room air. Sleep metabolic rate (SMR), substrate oxidation and sleep respiration quotient (RQ) were measured using direct calorimetry equipment (Omnical, Maastricht, the Netherlands). SMR was calculated as the lowest average 3-h energy expenditure during the sleep. At 6 AM the next morning, participants were woken up and left the respiration chamber.

**Blood parameters.** Venous blood samples were taken throughout the study in which routine medical laboratory analysis were performed (Tables 1 and 2). The metabolites phenylbutyrate, BCAA, BCKA and 3-HIB were analyzed in plasma by LC-MS, as previously described[23].

To extract metabolites from serum samples, 100 μl − 20° 40:40:20 methanol:acetonitrile:water (extraction solvent) was added to 5 μl of serum sample and incubated in −20 °C for 1 hour, followed by vortexing and centrifugation at 16,000 × g for 10 min at 4 °C. The supernatant (first extract) was transferred to a new tube. Then, 50 μl extraction solution was added to resuspend the pellet, followed by vortexing and centrifugation at 16,000 × g for 10 min at 4 °C. The supernatant (second extract) was combined with the first extract. Then, 3 μl among the 150 μl extract was loaded to LC-MS. A quadrupole-orbitrap mass spectrometer (Q Exactive, Thermo Fisher Scientific, San Jose, CA) operating in negative or positive ion mode was coupled to hydrophilic interaction chromatography via electrospray ionization and used to scan from m/z 70 to 1000 at 1 Hz and 75,000 resolution. LC separation was on a XBridge BEH Amide column (2.1 mm × 150 mm, 2.5 μm particle size, 130 Å pore size; Waters, Milford, MA) using a gradient of solvent A (20 mM ammonium acetate, 20 mM ammonium hydroxide in 95:5 water: acetonitrile, pH 9.45) and solvent B (acetonitrile). Data were analyzed using the MAVEN software[55]. Isotope labeling was corrected for natural $^{13}C$ abundance[56]. Flow rate was 150 μl/min. The LC gradient was: 0 min, 85% B; 2 min, 85% B; 3 min, 80% B; 5 min, 80% B; 6 min, 75% B; 7 min, 75% B; 8 min, 70% B; 9 min, 70% B; 10 min, 50% B; 12 min, 50% B; 13 min, 25% B; 16 min, 25% B; 18 min, 0% B; 23 min, 0% B; 24 min, 85% B; 30 min, 85% B. Autosampler temperature is 5 °C, and injection volume is 3 μL.

**Body composition.** On day 14 of each intervention period, participants were advised to have a lunch at 12:00 and to remain fasted until they arrived the research unit. In the afternoon, participants underwent a body composition measurement with the BodPod ® (Cosmed, California, USA). Body mass and body volume were assessed as previously described[57].

**Statistical analysis.** All results were normally distributed and presented as mean ± SE. The intervention effect was analyzed using the paired student $t$-test and correlations by using Pearson's correlation coefficient. Statistics were performed using SPSS 26.0 for Mac and a two-sided $p < 0.05$ was considered statistically significant.

**Reporting summary.** Further information on research design is available in the Nature Research Reporting Summary linked to this article.

## Data availability

The dataset generated during and analyzed during the current study are not publicly available. The corresponding author (E.P.) is the custodian of the data and will provide access to de-identified and processed participant data for academic purposes on request (esther.phielix@maastrichtuniversity.nl), with the completion of a data access agreement. The source data underlying Figs. 3 and 4, and Supplementary Fig. 1 are provided as a Source Data file. Source data and study protocol are provided with this paper immediately following publication with no end date. Source data are provided with this paper.

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

## Acknowledgements
E.P. was granted with a senior fellowship from the Dutch Diabetes Foundation (Grant No. 2017.82.010), which supported the present study. E.P. furthermore received a "VENI" Research Grant for innovative research from the Netherlands Organization for Scientific Research (91613132), an EFSD/Lilly grant from the European Foundation for the Study of Diabetes (EFSD). Z.A. is supported by NIH (DK114103). T.W. received a junior fellowship from the Dutch Diabetes Foundation (Grant No. 2015.81.1833). V.S. was supported by an ERC starting grant (Grant no. 759161) 'MRS in diabetes'. Metabolomics were performed with support from the DRC Regional Metabolomics and Fluxomics Core (NIH 5P30DK019525-45 7239). The funding agencies had no role in study design, data collection and analysis or manuscript writing.

## Author contributions
F.V., E.E.T., M.C.B. and M.N. performed the experiments and analyzed data. F.V. and E.P. wrote the manuscript. E.P., Z.A. J.H., P.S., T.W., M.C.M., V.S. and M.H. assisted during the acquisition, analysis and interpretation of data and reviewed the manuscript. E.P., P.S. and M.H. designed the study. All authors reviewed and approved the final version of the manuscript. E.P. is the guarantor of this work and, as such, had full access to all the data in the study and takes responsibility for the integrity of the data and the accuracy of the data analysis.

## Competing interests
The authors declare no competing interests.
