## [Peer Review File · Nature Communications]

Title: A randomized placebo-controlled clinical trial for pharmacological activation of BCAA catabolism in patients with type 2 diabetesREVIEWER COMMENTS

Reviewer #1 (Remarks to the Author):

In this revised manuscript, Vanweert et al perform a crossover study demonstrating that enhancing BCAA catabolism using NaPB for two weeks results in a modest improvement in insulin sensitivity in people with type 2 diabetes. The authors acknowledge that the study is limited by its small number (including small number of women) which does not allow controlling for medication status, sex, etc and short duration, but it provides proof of concept evidence that BCAA targeting with NaPB may deserve attention as a therapeutic modality. Acknowledging these limitations, I believe the authors have adequately addressed my comments with textual revisions.

Reviewer #2 (Remarks to the Author):

The manuscript has potential interest to the extent that it directly addresses questions related to the translatability of published studies of the pharmacodynamic effects of phenylbutyrate and the role of BCAA. The significance and interpretability are limited by the small size of the study and the small magnitude of the effects. The new analysis suggests that effects of phenylbutyrate were not observed in patients receiving metformin monotherapy but they only when metformin and sulfonylureas were co-administered. If this observation turns out to be reproducible, this could have implications for the underlying mechanisms.

Reviewer #3 (Remarks to the Author):

The authors have addressed some of my concerns, and have discussed limitations in the discussion section.

Reviewer #4 (Remarks to the Author):

1. Table 2 reports the changes in FPG observed in each arm (i.e., subtracting the time zero level from the level at the end of the 2-week treatment period), which is great. It might be also helpful to find out whether placebo alone can have significant decrease impact in glucose on the patients after 14 days of "placebo treatment". In the manuscript line 145-147, it mentioned "In addition, fasting glucose levels tended to be lower after NaPB treatment compared to placebo (7.7 ± 0.4 mmol/L vs. 8.2 ± 0.5 mmol/L, $p=0.06$, Table 2)". It can be a little misleading for the readers. Because in the table 2, there is no statistically significant difference in the change of glucose between NaPB and placebo (p value=0.09). This means that the placebo effect might also decrease the glucose level.

2. "All results were normally distributed are presented as mean \pm SE. The intervention effect was analyzed using the paired student t-test and correlations by using Pearson's correlation coefficient.":

- Have the authors measured and tested for deviation from normality? Since the student t-test relies on a normality assumption, otherwise, more conservative non-parametric tests should be used.
- The authors should consider using linear mixed-effects model given that each participant was tracked over time and had multiple visit/blood sample, especially when the study implemented cross-over of the treatments.

REVIEWER COMMENTS

We would like to thank the reviewers for their thoughtful comments and effort towards improving our manuscript. The reviewers found that our previous revision improved the manuscript.

A statistical referee (reviewer 4) mentioned some additional points, which we addressed below.

Additionally, we revised our Abstract following the CONSORT recommendations and identified the pre-specified primary and secondary outcomes in the method section (highlighted in yellow).

Reviewer #4

1. Table 2 reports the changes in FPG observed in each arm (i.e., subtracting the time zero level from the level at the end of the 2-week treatment period), which is great. It might be also helpful to find out whether placebo alone can have significant decrease impact in glucose on the patients after 14 days of “placebo treatment”. In the manuscript line 145-147, it mentioned “In addition, fasting glucose levels tended to be lower after NaPB treatment compared to placebo (7.7 ± 0.4 mmol/L vs. 8.2 ± 0.5 mmol/L, $p=0.06$, Table 2)”. It can be a little misleading for the readers. Because in the table 2, there is no statistically significant difference in the change of glucose between NaPB and placebo (p value=0.09). This means that the placebo effect might also decrease the glucose level.

We thank the reviewer for raising this point. The placebo did not decrease FPG after 2-week placebo treatment (8.5 ± 0.5 mmol/L vs. 8.2 ± 0.5 mmol/L, $p=0.30$), and therefore we can state that placebo did no impact glucose in the participants after the treatment. By design, for all other parameters measured, only end-of-treatment data were reported and compared between intervention arms. However, the FPG levels were measured before the treatment arms due to safety reasons and we reported their baseline values upon a reviewers' request.

2. “All results were normally distributed are presented as mean \pm SE. The intervention effect was analyzed using the paired student t-test and correlations by using Pearson’s correlation coefficient.”: •Have the authors measured and tested for deviation from normality? Since the student t-test relies on a normality assumption, otherwise, more conservative non-parametric tests should be used.

We want to thank the reviewer for this question. Indeed, the Shapiro-Wilk normality test was used to evaluate if data was normally distributed. All variables had a normal distribution, and therefore the parametric paired student t-test was used.

•The authors should consider using linear mixed-effects model given that each participant was tracked over time and had multiple visit/blood sample, especially when the study implemented cross-over of the treatments.

We thank the reviewer for this comment. As addressed with point 1, by design, patients were not tracked over time. Parameters were only measured at the end of treatment, and compared to placebo. Due to the invasive nature of the methodology applied (whole-body insulin sensitivity measured with an intravenous infusion test and mitochondrial oxidative capacity in muscle biopsies), we decided to perform a cross-over design, with our primary and secondary outcomes solely measured at the end of the 2-week treatment period. Therefore, we analysed all our results with the paired student t-test.

REVIEWERS' COMMENTS

Reviewer #4 (Remarks to the Author):

The authors have addressed my concerns, and have discussed limitations in the discussion section.